# Combination Therapies Targeting Apoptosis in Paediatric AML: *Understanding the Molecular Mechanisms of AML Treatments Using Phosphoproteomics*

**DOI:** 10.3390/ijms24065717

**Published:** 2023-03-16

**Authors:** Ahlam A. Ali, Lauren V. Cairns, Kathryn M. Clarke, Jaine K. Blayney, Katrina M. Lappin, Ken I. Mills

**Affiliations:** 1Wellcome Wolfson Institute for Experimental Medicine, School of Medicine, Dentistry and Biomedical Sciences, Queens University Belfast, Belfast BT9 7AE, Northern Ireland, UK; 2Patrick G Johnston Centre for Cancer Research, Queen’s University Belfast, Belfast BT9 7AE, Northern Ireland, UK; 3Haematology Department, C-Floor Tower Block, Belfast City Hospital, Belfast BT9 7AB, Northern Ireland, UK

**Keywords:** AML, apoptosis, paediatric, drug screening, synergism, double combination, triple combination, phosphoproteomics

## Abstract

Paediatric acute myeloid leukaemia (AML) continues to present treatment challenges, as no “standard approach” exists to treat those young patients reliably and safely. Combination therapies could become a viable treatment option for treating young patients with AML, allowing multiple pathways to be targeted. Our in silico analysis of AML patients highlighted “cell death and survival” as an aberrant, potentially targetable pathway in paediatric AML patients. Therefore, we aimed to identify novel combination therapies to target apoptosis. Our apoptotic drug screening resulted in the identification of one potential “novel” drug pairing, comprising the Bcl-2 inhibitor ABT-737 combined with the CDK inhibitor Purvalanol-A, as well as one triple combination of ABT-737 + AKT inhibitor + SU9516, which showed significant synergism in a series of paediatric AML cell lines. Using a phosphoproteomic approach to understand the apoptotic mechanism involved, proteins related to apoptotic cell death and cell survival were represented, in agreement with further results showing differentially expressed apoptotic proteins and their phosphorylated forms among combination treatments compared to single-agent treated cells such upregulation of BAX and its phosphorylated form (Thr167), dephosphorylation of BAD (Ser 112), and downregulation of MCL-1 and its phosphorylated form (Ser159/Thr 163). Total levels of Bcl-2 were decreased but correlated with increased levels of phosphorylated Bcl-2, which was consistent with our phosphoproteomic analysis predictions. Bcl-2 phosphorylation was regulated by extracellular-signal-regulated kinase (ERK) but not PP2A phosphatase. Although the mechanism linking to Bcl-2 phosphorylation remains to be determined, our findings provide first-hand insights on potential novel combination treatments for AML.

## 1. Introduction

Leukaemia is the most frequently (30%) occurring paediatric cancer. Of these, approximately 80% are acute lymphoblastic leukaemia (ALL), with acute myeloid leukaemia (AML) cases making up the remaining 20%, alongside other leukaemias. Management of ALL has improved dramatically in recent years, with around 90% of patients surviving for 5 years or more [1]. Unfortunately, children with AML do not have such a promising prognosis, with only 60% surviving 5 years or longer. This coupled with a high risk of relapse, the side effects of the harsh chemotherapy regimen, and a risk of not responding to induction therapy, means that the outlook is bleak. Such suboptimal outcomes are the result of numerous mutations and epigenetic changes occurring in this disease that adversely affect the susceptibility to treatment and relapse rate [2]. Therefore, novel treatment strategies are urgently needed to optimise outcome and minimise side effects in children. Recent data strongly indicate the need for age-specific therapies for AML patients, with paediatric AML cases having a different mutational landscape compared with AML diagnosed in adult patients [3].

The combination of two or more therapeutic treatments to specifically target cancer-inducing or cell-sustaining pathways is a cornerstone of cancer therapy [4]. A very promising emerging strategy is to interfere with the cells’ capability to evade programmed cell death pathways, such as apoptosis. Apoptosis is recognised as being dysregulated in AML, and overexpression of many anti-apoptotic proteins, such as B-cell lymphoma 2 (Bcl-2), BCL-XL, and myeloid cell leukaemia-1 (MCL-1), has been associated with poor outcomes in AML [5]. Consequently, many therapeutic agents aim to target factors that either upregulate pro-apoptotic factors or inhibit pro-survival mechanisms. One example of this is the anti-apoptotic Bcl-2 inhibitor Venetoclax. Preclinical data demonstrated the anti-leukaemic efficacy of Venetoclax in AML and its synergy when combined with hypomethylating agents or conventional chemotherapy agents [6], and clinical trials have confirmed the clinical benefit of Venetoclax-based therapies in newly diagnosed AML, leading to FDA breakthrough designations for these treatment combinations [7]. Despite promising results of clinical trials investigating new combination therapies, further clinical studies involving greater numbers of paediatric patients are still needed to improve the outcomes in childhood AML. 

Recently, published data generated from the Therapeutically Applicable Research to Generate Effective Treatments (TARGET) AML initiative [8], a collaborative Children’s Oncology Group (COG)–National Cancer Institute (NCI) project to characterise the mutational, transcriptional, and epigenetic landscape of childhood AML, have significantly broadened our understanding of the biology of AML in children and how it differs from adults [3]. Using the TARGET AML transcriptomic data, we previously identified gene expression anomalies in cell death and survival pathways, particularly amongst cytogenetically normal AML patients [9]. The interactions of the genes provide evidence that the apoptotic pathways are deregulated. On this basis, our group previously screened 83 apoptotic inducing compounds as single agents against paediatric AML cell lines. Those cell lines covered a spectrum of cytogenetic variations and mutational backgrounds and included the Kasumi-1 cell line (from a 7-year-old boy with AML FAB M2 (in the second relapse after bone marrow transplantation) with the t(8;21) translocation and a KIT mutation N822), MV4-11 cell line (from a 10-year-old boy with AML FAB M5 and t(4;11) translocation and an FLT3 internal tandem duplication), and the CMK cell line (originating from a 10-month-old boy with Down’s syndrome with AML M7 at relapse) [9]. We wanted to investigate if targeting different apoptotic pathways through combinations may be more beneficial than single treatments. Using an algorithm with the capability of determining all possible pairwise combinations for a given number of compounds, Lappin et al. (2020) showed the synergism of 10 compounds (ABT-737, Nutlin-3, RG-7112, MG-101, SU9516, MI-773, SecinH3, LY2409881, Akti-1/2, and Purvalanol-A) in well-158 in both the MV4-11 and the CMK cell lines. None of the 10 single agents in this well had a substantial effect on cell viability, but two or more compounds in combination resulted in cell death [9]. 

Another potential 10 compound (Purvalanol-A, Nutlin-3, MI-773, Poziotinib, SecinH3, Akti-1/2, LY2409881, ABT-737, SU9516, and Embelin) hit well (155) from our previous study showed synergism (data not published), which was worth taking forward for deconvolution in this study. 

## 2. Results and Discussion

### 2.1. Deconvolution of 10 Drugs Identified One Consistent Double and Triple Combination across Multiple Paediatric Cell Lines as Potential Therapies for Paediatric AML

We previously identified a potential synergistic mechanism between two or more of the following 10 apoptotic compounds: Purvalanol-A, Nutlin-3, MI-773, Poziotinib, SecinH3, Akti-1/2, LY2409881, ABT-737, SU9516, and Embelin, in the CMK and MV4-11 cell lines (data not published). Compounds in this group exert their apoptotic function via different mechanisms such as inhibition of anti-apoptotic protein Bcl-2 (ABT-737), inhibition of p53-murine double minute 2 (MDM2) interaction (Nutlin-3 and MI-773), targeting epidermal growth factor receptor (EGFR) such as Poziotinib and SecinH3, modulating the PI3K/AKT signalling pathway (Akti-1/2, LY2409881, and Embelin), or compounds that inhibit cyclin-dependent kinases (CDK) such as Purvalanol-A and SU9516.

To identify a potential synergistic combination, the 10-compound combination was de-convoluted into 45 double combinations and 120 triple combinations. The cytotoxicity of these combinations was assessed over a period of 72 h at two final concentrations (0.1 μM and 0.5 μM). Using seven paediatric AML cell lines (Kasumi-1, CMK, MV4-11, MOLM-13, CMS, THP1, and PL-21), it was clear that each of the ten compounds had minimal or no effect as single agents on the viability of any cell line; however, a combination of two or more of the compounds resulted in a substantial increase in cell death. This was demonstrated by calculating the Z-score (increased loss of membrane integrity), which showed scoring < 2 as single agents, while the 10-compound combination demonstrated a Z-score ≥ 2 in all the tested cell lines expect THP1 (z-score = 1.42) and PL-21 (z-score = 1.59) (Appendix A). When comparing FLT3 mutations or MLL translocation profiles, it appeared that THP1 had -*FLT3*/+*MLLr*, while PL-21 had +*FLT3*/-*MLLr*. All other tested cell lines were either positive (MV4-11 and MOLM-13) or negative (Kasumi-1, CMK, and CMS) for both *FLT3* and *MLLr*. This suggests that co-existing +*FLT3*/+*MLLr* or its absence considerably impact the combination treatment. The recently published molecular classifications of AML [3] would suggest that molecular profiling may guide therapy for patients with AML. Recent examples including the development of FLT3 or IDH1-2 inhibitors support this theory [10,11]. However, a large number of patients harbour non-targetable genomic changes. Therefore, an alternative to this could be strategies targeting apoptosis that would be active across molecular classes of the disease, regardless of mutational and/or cytogenetic profiles.

It was notable that MV4-11 cells (+FLT3/+*MLLr*) had the highest Z-score (4.42), making it the most sensitive cell line to the combination treatment (Appendix A). Deconvolution into 45 pairwise combinations demonstrated variability in each cell line; for example, the Nutlin-3 + Poziotinib double combination was only effective in the Kasumi-1 cell line at both concentrations with Z-scores 2.2 and 2.5 at two final concentrations of 0.1 μM and 0.5 μM, respectively. However, only two double combinations were identified as potential hits across the three cell lines tested (Kasumi-1, MV4-11, and CMK). Both ABT-737 + Purvalanol-A and ABT-737 + LY2409881 were the highest-scoring among other pairs at both tested concentrations (0.1 μM and 0.5 μM) (Figure 1A). Similarly, all possible 120 triple combinations for the 10-compound hit combination were investigated. There was a clear variability among the three cell lines tested, but only two of the triple combinations showed consistent effectiveness in the three cell lines tested. Triple combinations of ABT-737 + AKTi-1/2+SU9516 and ABT-737 + LY2409881+ SU9516 generated Z-scores > 2 across the three tested cell lines (Figure 1B). We decided to drop any combination that included LY2409881 as it showed higher cytotoxicity as a single agent in comparison with the other single agents, although LY2409881 had a Z-score below the cut-off of two, but it was high enough to likely to skew the findings of the combination treatments.

ABT-737 + Purvalanol-A and ABT-737 + AKTi-1/2 + SU9516 combinations were taken forward for validation and further analysis. We previously used the same strategy to deconvolute double combinations from a potential hit with similar set of 10 drugs (ABT-737, Nutlin-3, RG-7112, MG-101, SU9516, MI-773, SecinH3, LY2409881, Akti-1/2, and Purvalanol-A), and we found ABT-737 + Purvalanol-A to be effective in cell lines with a *FLT3-ITD* and an *MLL* rearrangement including the MV4-11 and MOLM-13 models [9], which confirms our findings. It is worth mentioning that the potential triple combination of ABT-737+AKTi-1/2 + SU9516 was further de-convoluted into its three double combinations: ABT-737 + AKTi-1/2, AKTi-1/2 + SU9516, and ABT-737 + SU9516 and tested across the five cell lines (Kasumi-1, CMK, MV4-11, MOLM-13, and CMS) at two different doses (0.1 μM, 0.5 μM) in order to ensure the same effect could not be obtained with only two of the compounds. The results showed no significant cytotoxicity for each of the double combinations compared to the single agents (Figure 1C), which confirms the efficiency of ABT-737 + AKTi-1/2 + SU9516 triple combination. 

### 2.2. Validation of Double and Triple Combinations: ABT-737 + Purvalanol-A and ABT-737 + AKTi-1/2 + SU9516 

Our double and triple combinations were further validated using the CellTiter^®^ Glo proliferation assay. Following 72 h incubation, the assay indicated that both double and triple combinations demonstrated a statistically significant decrease in cell viability compared to the single agents across Kasumi-1, MOLM-13, CMK, CMS, and MV4-11 cell lines, but not in THP1 or PL-21 cells, validating our screen results (Figure 2A).

In order to assess the efficacy of the apoptosis targeted drug combinations, ABT-737 + Purvalanol-A and ABT-737 + AKTi-1/2 + SU9516, we explored varying concentrations and dosing schedules across multiple paediatric AML cell lines for optimal synergy. Cell viability was assessed using the CellTiter-Glo^®^ assay over a time course of 72 h (Figure 2A). Determination of a synergistic relationship using the combination index (CI) values was performed using the Chou and Talalay method [12]. Dose–response curves for each of the single agents were used to identify the fraction affected at multiple concentrations (Appendix A), and this was compared to the fraction affected following treatment with the combinations using CompuSyn software-version 1 (ComboSyn Inc. Paramus, NJ, USA). CI value > 1 indicates an antagonistic response, CI value = 1 indicates an additive effect, and CI value < 1 is indicative of a synergistic effect. Among the cell lines tested (Kasumi-1, CMK, MV4-11, MOLM-13, and CMS), three cell lines—Kasumi-1, MOLM-13, and MV4-11—demonstrated a synergistic relationship when treated with the double combination of ABT-737 + Purvalanol-A, with CI values < 1; MV4-11 cells showed synergism with CI values < 1, but only when the concentration of ABT-737 was adjusted to a lower dose of 10 nM (Figure 2B). Furthermore, we examined the relationship of ABT-737 + AKTi-1/2 + SU9516 triple combination, which showed a CI value < 1 in Kasumi-1, MOLM-13, and MV4-11 cell lines, but again only at lower doses of ABT-737 (10 nM) in MV4-11 cells (Figure 2B). For further mechanistic validation, we chose to validate with only two cell lines, namely, Kasumi-1 (-*FLT3-ITD* and *MLLr*) and MV4-11 (+*FLT3-ITD* and *MLLr*) cells. 

Apoptosis was quantified using annexin V/PI flow cytometry following 72 h treatment with our double and triple combinations alongside single agent treatments. Flow cytometric analysis showed that both Kasumi-1 and MV4-11 cell lines demonstrated a statistically significant increase in annexin V-positive cells in the combination compared to the single agent treatments (Figure 3A). Quantification showed that 53.6% and 66% of Kasumi-1 cell line were apoptotic following treatment with ABT-737 + Purvalanol-A and ABT-737 + AKTi-1/2 + SU9516, respectively, compared to single treatments, which highlighted that approximately 15% of cells were undergoing apoptosis. The effect was higher in the MV4-11 cell line, with 69.87% and 84.5% being apoptotic following treatment with of ABT-737 + Purvalanol-A and ABT-737 + AKTi-1/2+SU9516, respectively, compared to single agent treatment (20% apoptotic).

Consistent with the annexin V/PI flow cytometry findings, the same combination concentrations and exposure times of ABT-737 + Purvalanol-A and ABT-737 + AKTi-1/2 + SU9516 resulted in cleavage/activation of caspase-9 and -3, and degradation of poly-ADP-ribose polymerase (PARP) compared to the single agents (Figure 3B). Densitometry analysis was performed by comparing the relative density of each sample following normalisation to GAPDH loading control (Appendix A). Enhanced expression of cleaved caspase-9 and -3 together with cleaved PARP suggest intrinsic activation of apoptosis. This was mediated through reactive oxygen species (ROS) as we showed a significant accumulation of ROS (Figure 3C). Previous work supports the fact that the antiproliferative effect of cyclin-dependent kinase (CDK) inhibitors is linked to the induction of oxidative stress [13]. CDK inhibition may be particularly successful in haematologic malignancies, which are more sensitive to inhibition of cell cycling and apoptosis induction. Indeed, the cell cycle is an interesting molecular target in AML, and CDK inhibitors are currently being studied in this respect especially 2, 4, and 9 [14]. Interestingly, two apoptotic compounds in our combination therapies are the CKD inhibitors Purvalanol-A (CDK2, -4, -9) and SU9516 (CDK2). Purvalanol-A has demonstrated induction of apoptosis against various cancer cells in vitro, either alone or in combination. SU9516 represents a group of very specific CDK inhibitors that are able to stop the cell cycle by inhibiting CDK1 and -2. SU9516 was previously demonstrated to be an effective therapeutic target in AML via CDK2 inhibition. The study showed that SU9516 inhibited cell proliferation without inducing cytotoxicity in human hematopoietic stem cells [15]. CDK inhibitors were also shown to downregulate MCL-1 transcription [16], an anti-apoptotic member of the BcL-2 family proteins, which is discussed later.

### 2.3. Profiling the Apoptotic Drug Combination Responsive Phosphoproteome in AML Cell Lines

Protein phosphorylation is an essential post-translational modification in most cellular processes. Proteomic and phosphoproteomic patterns associated with prognosis of AML patients and its progression from diagnosis to chemo-resistant relapse has been recently described [17]. To expand our understanding of the apoptotic mechanism involved, we applied a quantitative phosphoproteomic approach using tandem mass spectrometry (Figure 4A) to monitor dynamic changes of phosphorylation states in Kasumi-1 and MV4-11 cell lines following combination treatments. Functional enrichment, network analysis, and database mining were performed to identify biological processes and signalling pathways that are responsive to combination treatments of cells treated with ABT-737 + Purvalanol-A or ABT-737 + AKTi-1/2 + SU9516 (all drugs at 500 nM) for Kasumi-1 and MV4-11 cell lines, with the only difference for the MV4-11 cell line being ABT-737 used at 10 nM for treatments, wherein doses used were based on the maximum synergistic response identified earlier. Single treatments and DMSO controls were also tested. Using this experimental approach, phosphoproteomic analysis of Kasumi-1 led to the identification of 1646 phosphosites corresponding to 1464 unique phosphopeptides (90% pS, 9.8% pT, and 0.2% pY) on 897 proteins. Similarly, in MV4-11 cells, 1349 phosphosites were identified, corresponding to 1186 unique phosphopeptides (86.2% pS, 13.3% pT, and 0.5% pY) on 706 proteins. In parallel, proteomic analysis led to identification of 7554 and 6978 proteins in the Kasumi-1 and MV4-11 cell lines, respectively. A total of 628 phosphopeptides and 5631 proteins overlapped between both AML cell lines (Table 1).

Changes in phosphorylation and protein levels between controls and treated cells were assessed using Student’s *t*-test, and *p*-values adjusted for multiple comparisons of <0.05 were considered statistically significant. We also applied a fold-change (treated vs. control) threshold of 1.5 (FC ≥ 1.5) to define the down- and upregulated phosphopeptides and proteins. In our analysis, we decided to focus on apoptosis- and cell-death-related proteins in which 25 phosphoproteins and 139 proteins were significantly modulated in Kasumi-1 cells, and 28 phosphoproteins and 195 proteins in MV4-11 cells. Of note, most of the differentially modulated proteins were upregulated after combination treatment compared to single treatments (Figure 4B). Among the two cell lines tested were two proteins found to be lowly expressed but highly phosphorylated in combination treatments, whether it was the double or triple combination compared to single treatments and controls, namely, Bcl-2 protein and apoptosis regulator BAK. In addition, two apoptotic proteins were found to be normally expressed but poorly phosphorylated, namely, Bcl-2-associated agonist of cell death (BAD) protein and MCL-1. On the other hand, three proteins were found to be highly expressed and highly phosphorylated, namely, voltage-dependent anion-selective channel protein 1 (VDAC-1), TNF-receptor-associated factor 6 (TRAF6), and cellular FLICE-like inhibitory protein (c-FLIP); therefore, these proteins were brought forward for validation to uncover the synergistic mechanism of the combination treatments. 

### 2.4. Combination Treatments Induced Cell Death through Phosphorylation of Apoptotic Proteins 

Voltage-dependent anion channel (VDAC) proteins play a crucial role in controlling mitochondrial metabolism and apoptosis. VDACs are considered as pro-apoptotic proteins; therefore, therapeutic approaches aimed to counteract malignant cell proliferation have taken into account strategies directed to act on VDAC-1 expression. VDACs participate in the regulation of the channels responsible for ROS release into the cytosol [18]. VDAC-1 promotes the channel propensity to interact with the pro-apoptotic BAX and Bak proteins, which in turn blocks the ATP/ADP exchange [19]. BcL-2 facilitates mitochondrial respiration in AML cells, possibly through direct binding of mitochondrial pore protein VDAC-1. Inhibition of Bcl-2 disrupts BcL-2/VDAC interactions, reduces oxygen consumption rate, and induces ROS production [20]. Upregulation of VDAC-1 in combination treatments in our study confirms the intrinsic pathway of apoptosis, which could be via ROS production. The results also suggest that Bcl-2 inhibition by ABT-737 and high BAX levels might play a role in VDAC-mediated apoptosis (Figure 5A). VDAC-1 has been shown to be co-translationally or post-translationally modified by acetylation, phosphorylation, and/or S-nitrosylation, but the exact mechanism is unknown [21]. Proteomic data suggest it is likely to be phosphorylated, but there are no antibodies available to detect phosphorylated VDAC-1, which limited our investigation.

Both of our double and triple combination therapies had ABT-737, and like BH3-only proteins, it binds to anti-apoptotic Bcl-2 family members and antagonises their effects, thereby diminishing their ability to inhibit apoptosis. Suppression of pro-apoptotic BAX or BIM can lead to tumour formation and promote resistance to therapy in AML [22]. Our results showed upregulation of the apoptosis regulator BAX and its phosphorylated form (Thr167) following treatment with our combinations compared to single-agent treatments (Figure 5A), which suggests an activation role to enhance apoptosis in combination treatments. It has been reported that phosphorylation of BAX at serine S184 inactivates its pro-apoptotic function [23]. On the other hand, BAX phosphorylation at serine 163 or threonine 167 site leads to BAX activation and mitochondrial localisation prior to apoptosis [24]. Clinically, increased BAX expression is a good predictor of complete remission, being a pro-apoptotic marker (n = 34) in AML patients [25]. Consistent with other work, which showed that increased ROS plays a crucial role in BAX activation and cell death [26], we showed high levels of ROS production following our treatments, which might induce oxidative phosphorylation of BAX. Our combination therapies had ABT-737, which binds and inhibits Bcl-2, as well as directs BAX towards threonine 167 phosphorylation and therefore apoptosis. Indeed, Bcl-2 is believed to prevent BAX from releasing cytochrome c, thus restricting downstream activation of apoptotic machinery, and ABT-737 induces BAK/BAX-dependent apoptosis in AML cells [27]. Interestingly, the BAD protein was dephosphorylated on Ser 112 following ABT-737 + Purvalanol-A and ABT-737 + AKTi-1/2 + SU9516 combination treatments, but mostly in the triple combination treatment. A previous investigation highlighted an important role played by BAD phosphorylation for AML cell survival. The study showed that treatment with the serine/threonine kinase (AKT) inhibitor perifosine resulted in BAD dephosphorylation in THP-1 and MV4-11 cells [28]. The fact that our triple combination ABT-737 + AKTi-1/2 + SU9516 had an AKT inhibitor AKTi-1/2 supports this argument. Certainly, AKT, the downstream effector of phosphatidylinositol 3-kinase (PI3K), is known to play an important role in anti-apoptotic signalling and has been implicated in the aggressiveness of AML [28]. AKT was found to be constitutively phosphorylated in the majority of AML patients with high blast cell counts and was an adverse prognostic factor, and inhibition of AKT activity induces apoptosis of AML blasts [29]. Therefore, our combination treatment of ABT-737 + AKTi-1/2 + SU9516 had the advantage of blocking this pathway, which could potentially contribute to the synergistic effect. Estruch et al.’s (2020) study proposed that AML patients with the high mutational activity of PI3K/AKT signalling partially exhibit drug resistance through collateral inhibition of chemotherapy-induced apoptosis. This inhibition is potentially promoted by upregulation of the anti-apoptotic BcL-2 and MCL-1 proteins [30]. This was supported by preclinical PDX trials of AML and solid cancers supporting the therapeutic efficacy of the Bcl-2 inhibitor Venetoclax in combination with PI3K/AKT inhibitors [31]. Although numerous phase I/II clinical trials have tested PI3K/AKT pathway inhibitors in combination with current standard treatments for cancer patients, none of these inhibitors have been approved for single or combinatorial treatment of AML patients [32]. Therefore, there is a strong rationale to explore the therapeutic potential of our triple combination ABT-737 + AKTi-1/2 + SU9516 in AML.

In contrast to intrinsic apoptosis, the extrinsic apoptotic pathway relies on tumour necrosis factor (TNF) family death receptors for triggering apoptosis. Phosphoproteomic analysis showed both upregulation of TRAF6 and c-FLIP, together with their phosphorylated forms, suggesting activation of this pathway; however, caspase-8 assays (Figure 5B) showed no activation, which indicated that combination treatments induced apoptosis mainly via the intrinsic pathway in both AML models.

### 2.5. Combination Therapies Induce Phosphorylation of Anti-Apoptotic Bcl-2 That Drives Apoptosis 

Since phosphorylation is reversible and apoptosis is irreversible, it is possible that phosphorylation at some point facilitates irreversible damage to macromolecules, which in turn commits cells to death. It has been well established that both Bcl-2 and MCL-1 are frequently overexpressed in AML cells, and their overexpression is linked to poor prognoses and chemotherapy resistance [5]. Consistent with previous findings [6,7], our validation found that levels of Bcl-2 decreased following ABT-737 treatment compared to the control (Figure 5A). This is not surprising as it is well known for ABT-737, the BH3 mimetic, to have high potency activity against BcL-2. Furthermore, combination treatment resulted in even lower levels of Bcl-2 compared to single treatments and the control, which suggest that CDK and Akt inhibitors could potentially play a role in BcL-2 inhibition. Indeed, in a multiple myeloma model, CDK inhibitors potentiate pan-BH3 mimetic activity- obatoclax through a cooperative mechanism involving upregulation of Bim, Noxa, and Bik/NBK with coordinate downregulation of their anti-apoptotic counterparts including Bcl-2, Bcl-xL, and MCL-1 [33]. The fact that Akt upregulates Bcl-2 and MCL-1 [34] clearly explains the enhanced downregulation that we observed with the triple combination of ABT-737 + AKTi-1/2 + SU9516. 

All previous results strongly support the rationale of apoptotic combinational treatments. Studies have shown that AMLs harbouring RAS-MAPK-pathway-activating mutations (N/KRAS, FLT3, and PTPN11) demonstrate resistance to Venetoclax-based therapies, and a recent study demonstrated the benefit of combination therapies of Venetoclax with trametinib for RAS-mutated relapsed or refractory myeloid malignancies [35]. In fact, long-term exposure to single-agent ABT-737 in vitro can result in the emergence of resistant clones through upregulation of anti-apoptotic Bcl-2–related proteins [36], in addition to ABT-737 poor oral bioavailability [37], necessitating the urge for combinational strategies of those apoptotic drugs to overcome those issues. ABT-737 has low affinity to inhibit MCL-1, and therefore combining such agents with drugs capable of downregulating MCL-1 is a particularly appealing therapeutic strategy in AML. In this context, both SU9516, the CDK inhibitor, and AKTi-1/2, the AKT inhibitor, downregulate MCL-1, thus dramatically increasing ABT-737 lethality in AML cells. Although proteomic data suggest a normal level of MCL-1 expression, our validations demonstrated a significant down regulation of MCL-1 in combination treatments compared to all single treatments. This was also coupled with the downregulation of phosphorylated MCL-1 (Figure 5A). In support of this, it was found that cells that are initially sensitive to ABT-737 may become resistant by upregulating MCL-1 at transcriptional or post-translational levels. Consistently, ectopic or pharmacological modulation of MCL-1 protein levels restored sensitivity to ABT-737, thus indicating MCL-1 as an important determinant for ABT-737-induced cytotoxicity [38]. 

As shown with phospho-proteomic analysis, we thought that it was doubtful that Bcl-2 phosphorylation contributes to the enhanced synergism of combination treatments. To further explore our conjecture, protein expression investigations were performed. In theory, phosphorylated Bcl-2 was expected to be low given the ABT-737 effect on Bcl-2 inhibition; however, we saw increased levels of phosphorylated Bcl-2, which was consistent with our phospho-proteomic analysis predictions. It is well established that the BcL-2 protein undergoes multi-site phosphorylation localised at the loop region that includes Thr56, Thr69, Ser70, Thr74, and Ser87 in response to a diverse type of stimuli, which has been linked either to cell survival or to cell death. In AML, BcL-2 phosphorylation at Ser70 has been found necessary for full and potent anti-apoptotic function, and it has been associated with poor survival in AML [39]. However, in other cancer models, phosphorylation of serine 70 of Bcl-2 peaks after 48 h after paclitaxel treatment and accelerates apoptosis in breast cancer cells [40]. It has been suggested that such multi-site phosphorylation results in inhibition of Bcl2′s anti-phosphorylated sites at serine-70 and -87, as well as threonine-69. It might be that multi-site phosphorylation could drive protein degradation [39], but further studies need to be conducted to unravel this paradox mechanism. To prevent apoptosis, Bcl-2 forms a heterodimer with BAX, and therefore it is possible that hyper-phosphorylation of Bcl-2 drives apoptosis through the disruption of BAX–Bcl-2 association; our findings of enhanced BAX expression with combination treatments support this argument. Further investigations using immunoprecipitation assays will confirm whether cell death was associated with decreased BCL-2/BAX heterodimerisation.

### 2.6. Bcl-2 Phosphorylation and Modulation of Sensitivity to Apoptotic Treatments Was Regulated by Extracellular-Signal-Regulated Kinase (ERK) but Not PP2A Phosphatase in AML Cells

Phosphatase 2A (PP2A) is a tumour suppressor that inactivates multiple components of growth and survival signalling pathways required for tumorigenesis such as the Akt, MAPK, and Wnt signalling pathways. PP2A catalyses the selective removal of phosphate groups from protein serine and threonine residues. PP2A inactivation frequently occurs in several solid and non-solid tumours including AML, leading to sustained activation of survival pathways or inhibition of apoptotic pathways [41]. To further analyse the relevance of pBcl-2, particularly at Ser70, which appears to be the most common phosphorylation that is linked with AML survival in the sensitivity of AML cells to apoptotic treatments [22], we explored the effect of increasing Bcl-2 phosphorylation levels. As the protein phosphatase PP2A has proven to be responsible for Bcl-2 dephosphorylation, we tested whether its inhibition could affect the sensitivity of AML cells to apoptotic treatments. Okadaic acid is the most widely used inhibitor of serine/threonine protein phosphatases and, more importantly, at the low dose used in these experiments (1 nM), okadaic acid is considered a selective inhibitor of PP2A, as shown in Figure 6A [42]. Pre-treatment of Kasumi-1 and MV4–11 cells with okadaic acid before apoptotic treatments failed to rescue any of the single apoptotic drug treatments in both Kasumi-1 and MV4–11 cells but improved the effect of both the double and combination treatments (Figure 6B), suggesting that the observed apoptotic effect in with combination treatments might be through phosphorylation of certain apoptotic targets. However, Western blots revealed that PP2A inhibition did not enhance Bcl-2 phosphorylation at serine 70, excluding its role in Bcl-2 phosphorylation (Figure 6C). These results, taken together, further support the hypothesis that Bcl-2 phosphorylation at Ser70 may not modulate sensitivity of AML cells to apoptotic treatments.

On the other hand, as ERK1 and ERK2 have been identified as direct kinases of Bcl-2 phosphorylation [39], we next addressed whether the MEK1/ERK inhibitor PD98059 was able to inhibit Bcl-2 phosphorylation and thereby modulate the cytotoxic effect of our apoptotic treatments in AML cells. Treatment of Kasumi-1 and MV4–11 cells with 0.5 μM PD98059 for 24 h totally blocked the constitutive activation of phospho-ERK1/2 (Figure 7A). Importantly, ERK phosphorylation was only induced with single drug treatments, but not as a combination (Figure 7B), and the MEK1/ERK inhibitor PD98059 was able to suppress the induction of ERK phosphorylation for the single drugs. This suggests that phosphorylation of certain anti-apoptotic proteins protected cells against drug-induced apoptosis. To further examine if the inhibition of ERK1/2 would promote single apoptotic drug-induced cell death in AML cells, Kasumi-1 and MV4-11 cells were treated with increasing concentrations of either ABT-737, Purvanolol-A, AKT1/2, or SU9516 and the ERK/MEK inhibitor PD98059, alone and in combination, for 72 h, as shown in Figure 7C. PD98059 itself has minimal apoptotic activity but worked synergistically with the four single compounds (CIs under 0.5 in all cases) in both cell lines. ABT-737 showed the best synergism; when both drugs were used in combination, there was more than a twofold increase in apoptosis at the lowest concentration (10 nM) and sixfold increase in apoptosis at the highest concentration of 1 μM in Kasumi-1 cells; this effect was even higher in MV4-11 cells, which showed an eightfold increase in apoptosis at the highest concentration of 1 μM (Figure 7C), indicating that the synergistic use of MEK/ERK inhibitors and ABT-737 has the added advantage of blocking ERK’s role as a Bcl-2 kinase. Kasumi-1 and MV4–11 cells treated with PD98059 showed potent inhibition of Bcl-2 and MCL-1 phosphorylation, particularly for the ABT-737 treatment. This indicates that inhibition of Bcl-2 and MCL-1 phosphorylation restored sensitivity to ABT-737 as a single agent, but this effect was not seen with combination treatments that already had maximum synergism. These data clearly suggest that those single-agent treatments, mostly ABT-737, could be highly effective anti-leukaemia agents when used in combination to overcome resistance.

## 3. Materials and Methods 

### 3.1. Cell Culture

Seven different cell lines were used in this study to represent the heterogeneity of paediatric AML. For initial screening, MV4-11, Kasumi-1, and CMK cells were used. The MV4-11 cell line was established from the blast cells of a 10-year-old male that presented with biphenotypic B-myelomonocytic leukaemia at diagnosis (AML FAB M5) [43]. The Kasumi-1 cell line was derived from a 7-year-old boy with AML FAB M2 (acute myeloblastic leukaemia with maturation) in second relapse after bone marrow transplantation [44]. The CMK cell line was established from the peripheral blood of a 10-month-old boy with Down’s syndrome and acute megakaryocytic leukaemia (AML M7) at relapse in 1985. For validation studies, MOLM-13, THP-1, PL-21, and CMS cell lines were used, each representing a case of AML with different cytogenetic and mutational backgrounds. All cell lines were obtained from the Deutsche Sammlung von Mikroorganismen und Zellkulturen (DSMZ) (Braunschweig, Germany), except the CMS line, which was provided courtesy of Dr Yubin Ge (Detroit, MI, USA) [45]. All cell lines were maintained in Roswell Park Memorial Institute (RPMI) 1640 (Thermo Fisher Scientific, Leicestershire, UK) supplemented with either 10–20% foetal bovine serum (FBS; Thermo Fisher Scientific, Leicestershire, UK) (MV4-11) or 100 µg/mL penicillin–streptomycin (Thermo Fisher Scientific, Leicestershire, UK). Cell lines were authenticated using short tandem repeat (STR) profiling carried out by the suppliers, incubated at 37 °C under 5% CO_2_, and sub-cultured every 3–4 days to maintain exponential growth.

### 3.2. Reagents and Screening Assay

When specified, cells were pre-incubated for 1 h with the MEK1/ERK (extracellular-signal-regulated kinase) inhibitor PD98059 (Monmouth, NJ, USA) or the PP2A/PP1 inhibitor okadaic acid (Merck, Poole, UK) prior to adding apoptotic treatments. All combination drug screening treatments were performed using the Echo liquid handling technology (Labcyte). The compound screening assay was carried out in 384-well black optical bottom plates (Nunc, Science Warehouse Limited). Cells were seeded at a density of 2 × 10^4^ cells per well. Compounds were diluted with dimethyl sulfoxide (DMSO) (Merck, Poole, UK) and added to cells to generate the desired drug concentration. Cells treated with 0.1% DMSO were used as a vehicle control. Following treatment, cells were incubated for 72 h in a humidified incubator at 37 °C supplemented with 5% CO_2_. Cell toxicity was assessed at 72 h using CellTox™ Green Cytotoxicity Assay (Promega, Southampton, UK) as per the manufacturer’s instructions. The relative fluorescence unit (RFU) (Ex: 485 nm, Em: 520 nm) was measured using a Synergy HTX Multi-Mode Micro-Plate reader (Biotek, VT, USA).

Cell viability was assessed using the CellTiter-Glo (Promega, Southampton, UK) luminescent assay. Following relevant treatments for a 72 h incubation period, cell culture medium from each well was added to an equal volume of CellTiter-Glo reagent. The plate was mixed on an orbital shaker for 2 min to induce cell lysis, followed by a 20 min incubation at room temperature to stabilise the luminescent signal. Luminescence was measured using a Synergy HTX Multi-Mode Microplate reader (Biotek, Winooski, VT, USA).

### 3.3. Western Blot Analysis 

Following cell treatments, the whole cell lysates were collected after mixing cell pellets with 100 μL of radio immune precipitation assay buffer (RIPA). Protein sample concentration was determined by using the bicinchoninic acid protein assay kit (Merck, Poole, UK). Each sample (20 μg) was electrophoresed through a 4–12% sodium dodecyl (SDS)-polyacrylamide gel (Life Technologies, Renfrewshire, UK), transferred onto a nitrocellulose membrane (Hybond-C, Amersham; UK), and probed with apoptotic antibodies (Danvers, MA, USA). GAPDH was used as a loading control (Merck, Poole, UK). Levels of protein expression were assessed using Pierce ECL Western blotting detection kit (Thermo Scientific; Leicestershire, UK). Membranes were scanned using benchtop G:box (Syngene, Cambridge, UK), and density was calculated using the ImageJ program (http://rsbweb.nih.gov/ij/ (accessed on 17 November 2022) incorporating correction of loading controls.

### 3.4. Annexin V Flow Cytometry

Cells were seeded at 2 × 10^5^ in a 6-well plate and treated with appropriate apoptotic drugs. After the 72 h treatment period, the media and cells were transferred to a 15 mL tube on ice and centrifuged at 2000 rpm for 5 min at 4 °C. The supernatant was removed, and the cells were re-suspended in 0.5 mL of phosphate-buffered saline (PBS). Following this, a further 3 mL of PBS was added, and the cells were centrifuged at 2000 rpm for 5 min at 4 °C. The supernatant was removed, and the cells were re-suspended in 300 µL of binding buffer and transferred to a flow cytometry tube (BD Biosciences). Each sample was subsequently stained with 4 µL of annexin V (BD Biosciences) and 4 µL of propidium iodide (BD Biosciences). The samples were then left to incubate in the dark for 20 min at room temperature. Following the incubation period, 330 µL of 1x binding buffer was added to each sample. Flow cytometry was carried out on a BD FACS Calibur system using the BD CellQuest™ Pro analysis software, version 4.

### 3.5. Reactive Oxygen Species Detection

For detection of reactive oxygen species (ROS) levels in AML cells following either single or combination treatment for 72 h, cells were collected by centrifugation, washed with PBS, and stained using a total ROS detection kit (ENZO life science, Exeter, UK) for 30 min at 37 °C in the dark. Finally, the stained cells were analysed for fluorescence, measured using a Synergy HTX Multi-Mode Micro-Plate reader (Biotek, VT, USA). We used 5 mM H_2_O_2_ and 5 mM N-acetyl cysteine (NAC) anti-oxidant as positive and negative controls, respectively.

### 3.6. Caspase-8 Activity Assays 

A caspase-8 assay colorimetric kit was purchased from Abcam, UK. Following 72 h of relevant apoptotic treatments, either alone or in combination with AML cells, cells were collected by centrifugation, washed with PBS, and protein concentrations were quantified using the Pierce™ BCA Protein Assay Kit from ThermoFisher Scientific, Leicestershire, UK. The activity of caspase-8 was assessed according to the protocol provided with the Caspase 8 Assay Colorimetric Kit. Caspase-8 activity was presented as fold increase in activity of the corresponding control.

The caspase-8 activity luminescence assay (Promega, Southampton, UK) was performed per the manufacturer’s instructions. Briefly, to each well containing treated cells in 100 μL cell culture medium, 100° μL of caspase-8 reagent solution was added. The reagent solution consists of a lysis/activity buffer containing a pro-luminogenic caspase substrate and a thermostable luciferase. In the presence of the corresponding caspase, the substrate is cleaved, releasing aminoluciferin digested by the luciferase. The resulting luminescent signal is proportional to the measured caspase activity. The plate was shaken for 30 s on a plate mixer and afterwards incubated for 30 min at RT in the dark. The luminescence signal was detected using Synergy HTX Multi-Mode Micro-Plate reader (Biotek, VT, USA).

### 3.7. Caspase-Glo™ Assay

The Caspase-Glo™ Assay (Promega, Southampton, UK) quantifies both caspase-3 and -7 activation, which is indicative of apoptosis. Following treatments, 50 µL of caspase reagent was mixed in equal portion of cell suspension in 96-well white plates. Each plate was incubated at room temperature for 45 min on the orbital shaker (Stuart Scientific, Nottingham, UK). The luminescence signal was detected using Synergy HTX Multi-Mode Micro-Plate reader (Biotek, VT, USA).

### 3.8. Proteomic Analysis

TMT Labelling, high-pH reversed-phase chromatography, and phospho-peptide enrichment.

MV4-11 and Kasumi-1 cells (10^6^ cells per each condition, three biological replicates) were treated or not with single and combination treatments for 72 h. After being collected by centrifugation and washed with PBS, cells were re-suspended in lysis buffer containing 2% SDS and 50 mM DTT. Aliquots of 100 µg of each sample were digested with trypsin (2.5 µg trypsin; 37 °C, overnight), labelled with Tandem Mass Tag (TMT) ten plex reagents according to the manufacturer’s protocol (Thermo Fisher Scientific, Loughborough, UK), and the labelled samples were pooled.

For the total proteome analysis, an aliquot of 50 µg of the pooled sample was desalted using a SepPak cartridge according to the manufacturer’s instructions (Waters, Milford, MA, USA). Eluate from the SepPak cartridge was evaporated to dryness and resuspended in buffer A (20 mM ammonium hydroxide, pH 10) prior to fractionation by high-pH reversed-phase chromatography using an Ultimate 3000 liquid chromatography system (Thermo Fisher Scientific). In brief, the sample was loaded onto an XBridge BEH C18 Column (130Å, 3.5 µm, 2.1 mm × 150 mm, Waters, UK) in buffer A, and peptides were eluted with an increasing gradient of buffer B (20 mM ammonium hydroxide in acetonitrile, pH 10) from 0-95% over 60 min. The resulting fractions (15 in total) were evaporated to dryness and resuspended in 1% formic acid prior to analysis by nano-LC–MS/MS using an Orbitrap Fusion Lumos mass spectrometer (Thermo Fisher Scientific, Loughborough, UK).

For the Phospho proteome analysis, the remainder of the TMT-labelled pooled sample was also desalted using a SepPak cartridge (Waters, Milford, MA, USA). Eluate from the SepPak cartridge was evaporated to dryness and subjected to TiO_2_-based phosphopeptide enrichment according to the manufacturer’s instructions (Pierce). The flow-through and washes from the TiO_2_-based enrichment were then subjected to FeNTA-based phosphopeptide enrichment according to the manufacturer’s instructions (Pierce, Lancashire, UK). The phospho-enriched samples were again evaporated to dryness and then re-suspended in 1% formic acid prior to analysis by nano-LC–MS/MS using an Orbitrap Fusion Lumos mass spectrometer (Thermo Fisher Scientific, Loughborough, UK).

### 3.9. Nano-LC–Mass Spectrometry

High-pH RP fractions (Total proteome analysis) or the phospho-enriched fractions (Phospho-proteome analysis) were further fractionated using an Ultimate 3000 nano-LC system in line with an Orbitrap Fusion Lumos mass spectrometer (Thermo Scientific). In brief, peptides in 1% *(vol/vol)* formic acid were injected onto an Acclaim PepMap C18 nano-trap column (Thermo Fisher Scientific, Loughborough, UK). After washing with 0.5% *(vol/vol)* acetonitrile 0.1% *(vol/vol*), formic acid peptides were resolved on a 250 mm × 75 μm Acclaim PepMap C18 reverse-phase analytical column (Thermo Fisher Scientific, Loughborough, UK) over a 150 min organic gradient, using 7 gradient segments (1–6% solvent B over 1 min, 6–15% B over 58 min, 15–32% B over 58 min, 32–40% B over 5 min, 40–90% B over 1 min, held at 90% B for 6 min, and then reduced to 1% B over 1 min) with a flow rate of 300 nl min^−1^. Solvent A was 0.1% formic acid and Solvent B was aqueous 80% acetonitrile in 0.1% formic acid. Peptides were ionised by nano-electrospray ionisation at 2.0 kV using a stainless-steel emitter with an internal diameter of 30 μm (Thermo Scientific) and a capillary temperature of 300 °C. 

All spectra were acquired using an Orbitrap Fusion Lumos mass spectrometer controlled by Xcalibur 3.0 software Thermo Fisher Scientific, Loughborough, UK) and operated in data-dependent acquisition mode using an SPS-MS3 workflow. FTMS1 spectra were collected at a resolution of 120,000, with an automatic gain control (AGC) target of 200,000 and a max injection time of 50 ms. Precursors were filtered with an intensity threshold of 5000, according to charge state (to include charge states 2–7) and with monoisotopic peak determination set to Peptide. Previously interrogated precursors were excluded using a dynamic window (60 s +/–10 ppm). The MS2 precursors were isolated with a quadrupole isolation window of 0.7m/z. ITMS2 spectra were collected with an AGC target of 10,000, max injection time of 70 ms, and CID collision energy of 35%.

For FTMS3 analysis, the Orbitrap was operated at 50,000 resolution with an AGC target of 50,000 and a max injection time of 105 ms. Precursors were fragmented by high energy collision dissociation (HCD) at a normalised collision energy of 60% to ensure maximal TMT reporter ion yield. Synchronous precursor selection (SPS) was enabled to include up to 10 MS^2^ fragment ions in the FTMS3 scan.

### 3.10. Proteomic Data Analysis

The raw data files were processed and quantified using Proteome Discoverer software v2.1 Thermo Fisher Scientific, Loughborough, UK) and searched against the UniProt Human database (downloaded August 2020: 167,789 entries) using the SEQUEST HT algorithm. Peptide precursor mass tolerance was set at 10ppm, and MS/MS tolerance was set at 0.6 Da. Search criteria included oxidation of methionine (+15.995 Da), acetylation of the protein N-terminus (+42.011 Da), and methionine loss plus acetylation of the protein N-terminus (−89.03 Da) as variable modifications and carbamidomethylation of cysteine (+57.0214) and the addition of the TMT mass tag (+229.163) to peptide N-termini and lysine as fixed modifications. For the phospho-proteome analysis, phosphorylation of serine, threonine, and tyrosine (+79.966) was also included as a variable modification. Searches were performed with full tryptic digestion, and a maximum of 2 missed cleavages were allowed. The reverse database search option was enabled, and all data were filtered to satisfy false discovery rate (FDR) of 5%.

### 3.11. Statistical Analysis

For drug screening, the RFU value for each well was normalised to the average RFU value for the DMSO controls on that plate. In all experiments, the viability of control cells exceeds 80%. The mean and standard deviation were calculated for the normalised RFU values, and a Z-score was calculated, taking into account the mean and standard deviation of the whole plate. Combination index (CI) values were calculated by the Chou and Talalay method [12], using the CompuSyn Software, version 1 (ComboSyn, Inc.). CI values < 1 indicate synergy; CI values = 1 indicate an additive effect, and CI values > 1 indicate an antagonistic effect. All statistical analysis was performed using the GraphPad software, version 9 (Prism). Unpaired *t*-tests and two-way ANOVA tests at 95% confidence intervals were utilised.

## 4. Conclusions

Targeted therapies that are designed to induce apoptosis in leukaemic cells are currently some of the most promising anti-leukaemia strategies, but using those therapies in combination will add a synergistic value by targeting multiple apoptotic targets. 

In this study, we identified potential novel drug combinations in a paediatric AML setting. To our best understanding, the specific double drug combination of ABT-737 + Purvalanol-A has only been reported by our group for this indication [9]. In addition, we identified a potential triple combination of ABT-737 + AKTi-1/2 + SU9516, which works by targeting three different mechanisms more collaboratively to enhance cytotoxicity, namely, Bcl-2, AKT, and CDK inhibition.

We observed distinct phosphorylation signatures and protein expression profiles associated with response towards our combination treatments. Such findings provide fresh clues related to anti-leukaemic effect with combinational treatments. Although the mechanism linking to Bcl-2 phosphorylation remains to be elucidated, our preclinical results support the clinical evaluation of apoptotic combination treatments as a novel therapeutic strategy for paediatric AML.

Proteomics has come a long way since the early 2000s, and we now acknowledge that there were high levels of false-positive peptide-to-spectrum matches in the reported results. Although we had some agreements between the proteomic and the experimental data, our findings confirm the importance of combining both proteomic with functional validation to fully provide answers that cannot be obtained from the study of individual proteins or groups of proteins.

## Figures and Tables

**Figure 1 ijms-24-05717-f001:**
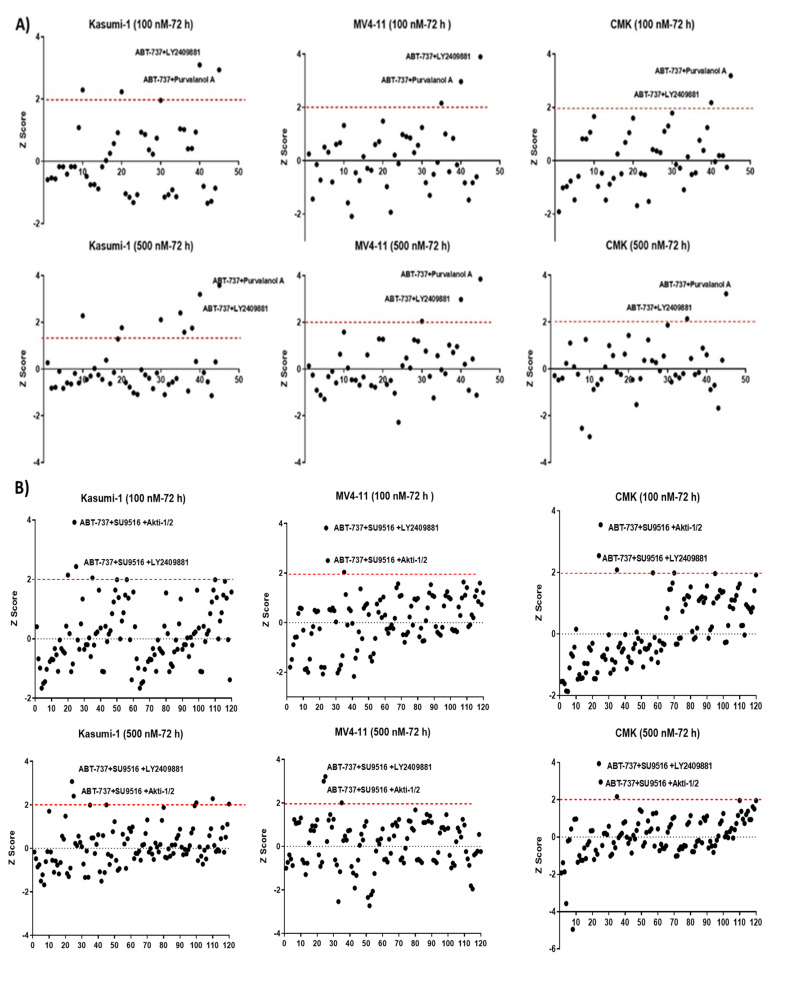
Secondary deconvolution screen of the 10 hit well into pairwise (**A**,**B**) triple combinations. The 10 compounds within this well were deconvoluted into 45 pairwise combinations and 120 triple combinations. The cytotoxicity of these combinations was tested over a period of 72 h at a final concentration of 0.1 and 0.5 μM in Kasumi-1, MV4-11, and CMK cells. Cells were treated with 0.1% DMSO as a vehicle control. Cell toxicity was assessed using the CellTox Green Cytotoxicity Assay, which was added to the cell suspension, and cells were incubated for 20 min at room temperature before reading. The relative fluorescence unit (RFU) (Ex: 485 nm, Em: 520 nm) was measured using a Synergy HTX Multi-mode Microplate reader. The graphs demonstrate the Z-score and are representative repeats of 3 independent screens. (**C**) Investigating the effects of the double combinations of the triple combination of ABT-737 + AKTi-1/2 + SU9516. The CellTox Green Cytotoxicity Assay was used to determine cell viability of the cell lines following 72 h treatment. The graphs demonstrate the Z-score and are representative repeats of 3 independent screens. * = *p* < 0.05.

**Figure 2 ijms-24-05717-f002:**
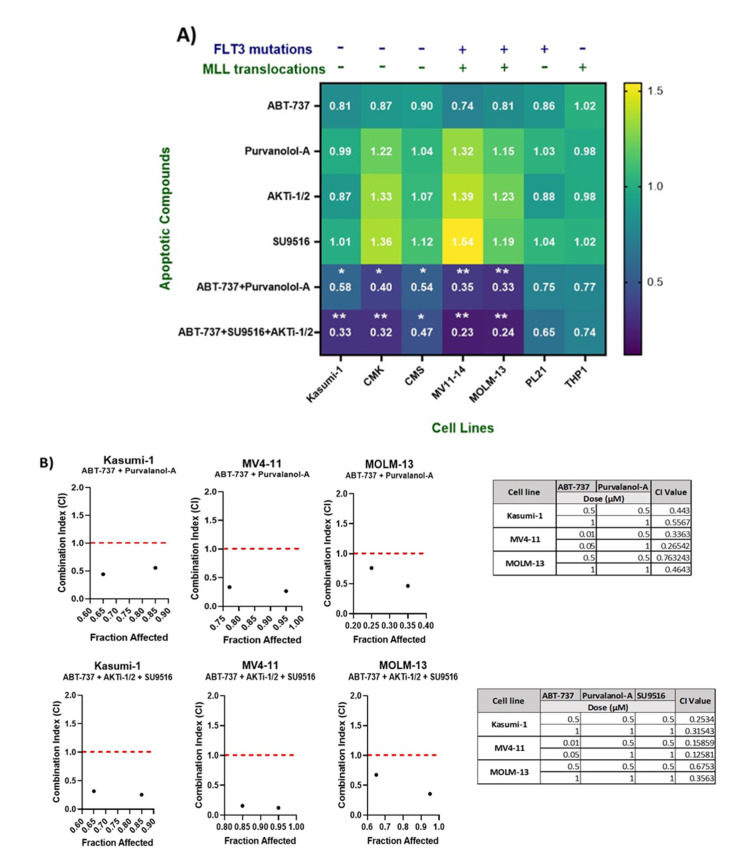
(**A**) Validation of combination treatments using CellTiter^®^ Glo, which measures ATP, which is proportional to cell viability after 72 h. Luminescence was measured using a Synergy HTX Multi-Mode Microplate reader. Values were normalised to DMSO controls. Data are representative of N = 3 ± SEM. (**B**) Optimisation of the best dose for the maximum synergistic effect of successful combinations using combination index (CI) values. Combination index values were determined for ABT-737 + Purvalanol-A and ABT-737 + AKTi-1/2 + SU9516 in Kasumi-1, MV4-11, and MOLM-13 cell lines. CI > 1 is antagonistic, CI = 1 is additive, and CI < 1 is synergistic. This combination demonstrated synergism at multiple concentrations across both cell lines. Data are representative of N = 3 ± SEM. * = *p* < 0.05, ** = *p* < 0.005.

**Figure 3 ijms-24-05717-f003:**
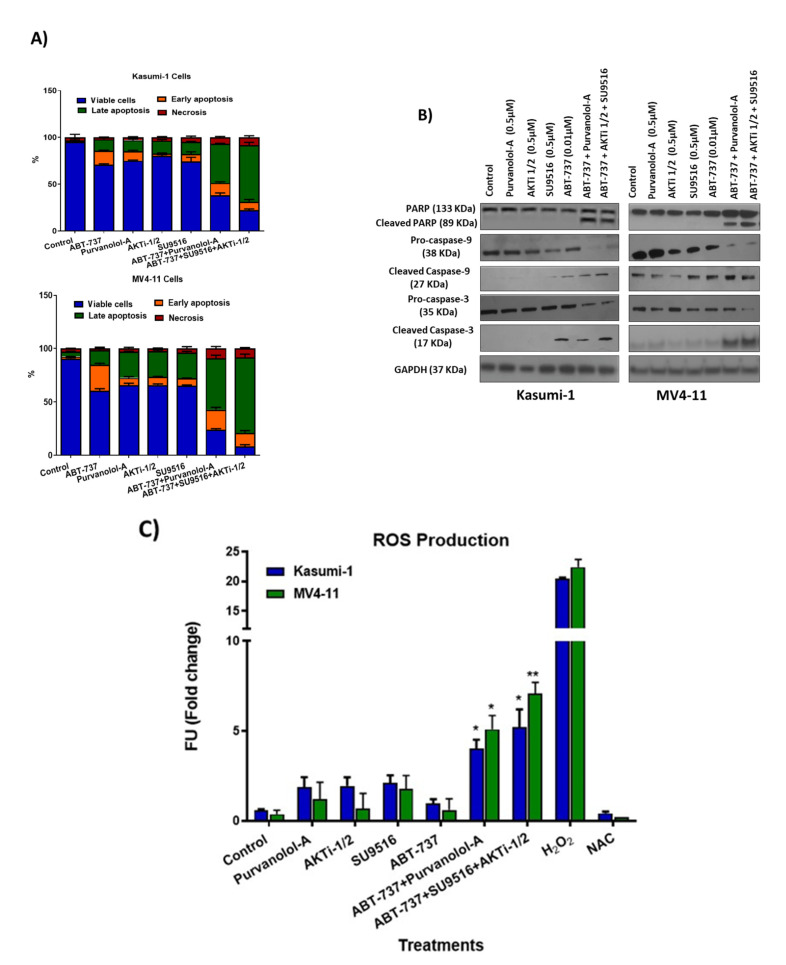
Validation of double and triple combinations in the paediatric AML cell lines. (**A**) Flow cytometry analysis of the annexin V-positive cell population for Kasumi-1 and MV4-11 cells following 72 h treatment as single agents and as a combination. (**B**) Western blot analysis of PARP, cleaved-PARP, procaspase-9, cleaved caspase-9, procaspase-3, and cleaved caspase-3 in Kasumi-1 and MV4-11 cell lines following 72 h treatment as single agents and as a combination. GAPDH was used as a loading control. All results shown are representative of three independent experiments. (**C**) Reactive oxygen species quantification following 72 h treatment as single agents and as a combination. Stained cells were analysed for fluorescence, measured using a Synergy HTX Multi-Mode Micro-Plate reader. We used 5 mM H2O2 and 5 mM N-acetyl cysteine (NAC) anti-oxidant as positive and negative controls respectively. * = *p* < 0.05, ** = *p* < 0.005 values of significance of treatments compared to controls were calculated using the paired *t*-test.

**Figure 4 ijms-24-05717-f004:**
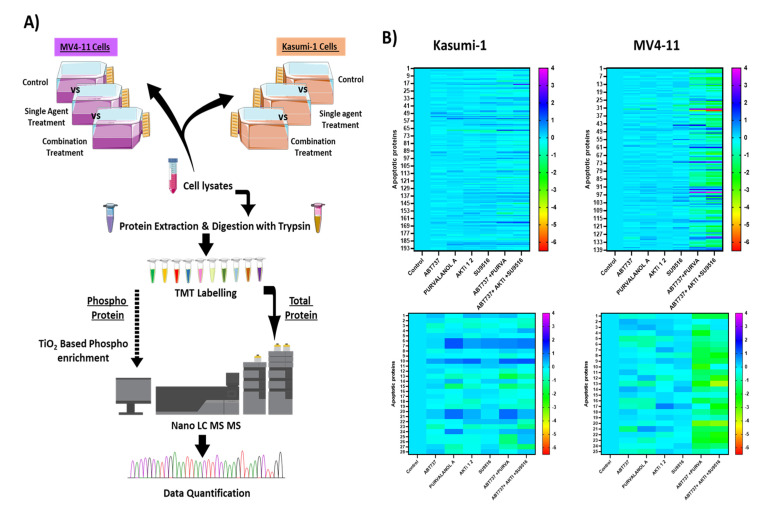
Phosphoproteomic and proteomic analysis of paediatric AML cells treated with apoptotic combination treatments: (**A**) Workflow for the investigation of phosphorylation changes induced in Kasumi-1 and MV4-11 cells after treatment with combination treatments of ABT-737 + Purvalanol-A or ABT-737 + AKTi-1/2 + SU9516. Three biological replicates of each group were evaluated. (**B**) Quantitative phosphoproteomic profiling of Kasumi-1 and MV4-11 cell identified proteins (**top**) and phosphoproteins (**bottom**) that were significantly modulated. Colour scale indicates fold-change (treated vs. control) threshold of 1.5 (FC ≥ 1.5) to define the down- and upregulated phosphopeptides and proteins.

**Figure 5 ijms-24-05717-f005:**
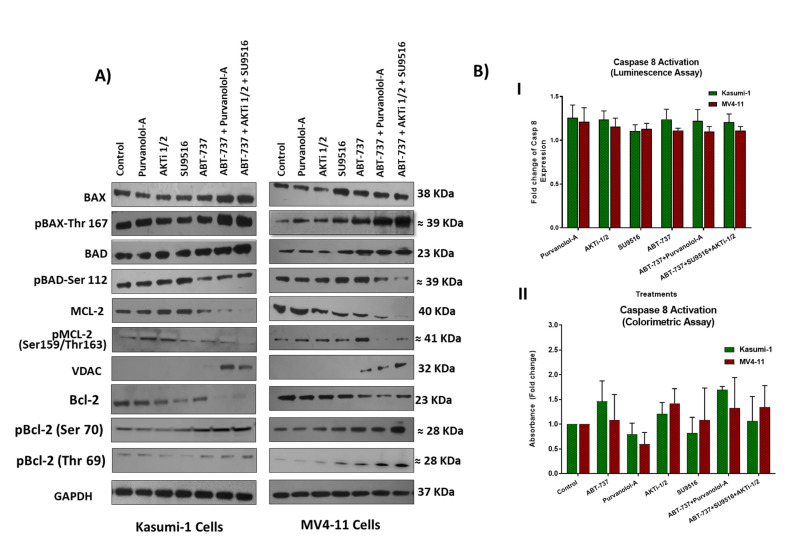
(**A**) Western blot analysis of pro-apoptotic and anti-apoptotic proteins and their phosphorylated versions in Kasumi-1 and MV4-11 cell lines following 72 h treatment as single agents and as a combination. GAPDH was used as a loading control. (**B**) Caspase-8 activity in cell lysates from Kasumi-1 and MV4-11 cells treated for 72 h as single agents and combination treatments of ABT-737 + Purvalanol-A and ABT-737 + AKTi-1/2 + SU9516 or single agents. Caspase activity was detected using either the caspase-8 assay colorimetric kit (**I**) where protein concentrations were quantified first, then the activity of caspase-8 was assessed, or using caspase-8 activity luminescence assay (**II**). The absorbance or luminescence signal was detected using Synergy HTX Multi-mode Micro-plate reader. Caspase-8 activity was presented as fold increase in activity of the corresponding control. All the results shown are representative of three independent experiments.

**Figure 6 ijms-24-05717-f006:**
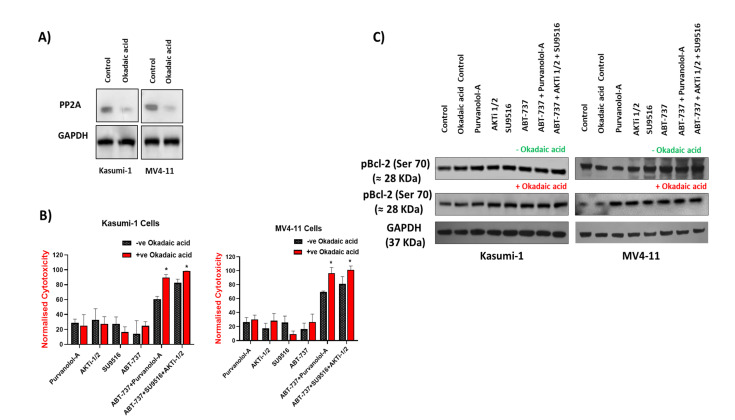
Bcl-2 phosphorylation was not regulated by PP2A phosphatase activities in AML cells. Kasumi-1 and MV4-11 cell lines were treated with 1 nM okadaic acid (OA) for 72 h, and total protein extracts were analysed for PP2A (**A**) or pBCL-2, BCL-2, and GAPDH expression (**C**). (**B**) Kasumi-1 and MV4-11 cell lines were co-treated with 1 nM OA and single/combination treatments for 72 h. Cells were treated with 0.1% DMSO as a vehicle control. Cell toxicity was assessed at 72 h using the CellTox Green Cytotoxicity Assay which was added to the cell suspension, and cells were incubated for 20 min at room temperature before reading. Relative fluorescence unit (RFU) (Ex: 485 nm, Em: 520 nm) was measured using a Synergy HTX Multi-Mode Microplate reader. Data were normalised to vehicle controls and are representative of N = 3 ± SEM. * = *p* < 0.05.

**Figure 7 ijms-24-05717-f007:**
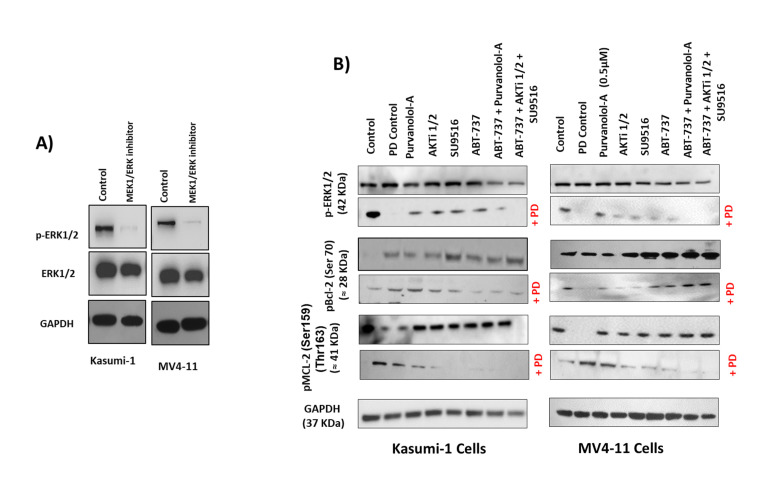
Bcl-2 phosphorylation was regulated by extracellular signal-regulated kinase (ERK). Kasumi-1 and MV4-11 cell lines were treated with 1 µM PD98059, and total protein extracts were analysed by Western blot for the expression of ERK/Perk (**A**) and pBCL-2(Ser70), pMCL-2 (Ser159/Thr163), and GAPDH expression (**B**). (**C**) Kasumi-1 and MV4-11 cell lines were co-treated with 1 µM PD98059 and increasing concentrations (10–1000 nM) of single drugs (ABT-737, Purvalanol-A, AKTi-1/2, and SU9516) for 72 h. Apoptosis was quantified using the Caspase-Glo™ Assay, which quantifies both caspase-3 and -7 activation, indicative of apoptosis. Following treatments, 50 µL of caspase reagent was mixed in equal portion with cell suspension in a 96-well white micotitre plate, incubated at room temperature for 45 min. The luminescence signal was detected using a Synergy HTX Multi-Mode Micro-Plate reader. Data were normalised to vehicle controls and are representative of N = 3 ± SEM.

**Table 1 ijms-24-05717-t001:** Number of identified and significantly modulated phosphopeptides and proteins in Kasumi-1 and MV4-11 cell lines.

Phosphoproteome and Proteome Dataset	Kasumi-1 Cells	MV4-11 Cells
Unique phosphosites	1646	1349
	Total: 2468	Overlap: 527
Unique phosphopeptides	1464	1186
	Total: 2022	Overlap: 628
Unique phosphoproteins	897	706
Unique phosphopeptides w/matching prot	1530	1267
Proteins (proteomic analysis)	7554	6978
	Total: 8901	Overlap: 5631

## Data Availability

The data that support the findings of this study are available from the corresponding author upon reasonable request.

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
