# Peer review of "Combination Therapies Targeting Apoptosis in Paediatric AML: Understanding the Molecular Mechanisms of AML Treatments Using Phosphoproteomics"

_ijms, 2023, doi:10.3390/ijms24065717_

Round 1
Reviewer 1 Report
The manuscript by Ali, et al, is a followup to an earlier study in which they reported on synergistic anti-AML activity for combinations of novel drugs that individually do not have activity. Here they systematically investigated the drug combinations to determine which were contributing to the anti-Aml activity. They then pursued an investigation of mechanism including proteomics, phosphoproteomics and functional assays. Both of the drug combinations that were pursued included ABT-737, also known as navitoclax, and so the mechanism questions came down to effects on BCL2. This is a nicely organized manuscript that is generally easy to follow, with conclusions supported by the data. Strengths of the paper include the multiple approaches to measuring anti-AML activity, and the validation of the proteomics results with immunoblotting and functional studies. A limitation of the study is that only cell lines were studied, and therefore the impact of the microenvironment and the impact of intratumor heterogeneity on efficacy cannot be determined. Another limitation is that changes in BCLxL were not shown or discussed, even though it too is a target of ABT-737. Other more minor suggestions are below:
1. Reference 1 seems to be about AML it is associated in the introduction with a statement about ALL. Perhaps a more relevant reference can be cited.
2. For Figure 2A, please state if the values in the heatmap are normalized to a control value, and if so, what the control condition was.
3. For the proteomics analysis, since there are a large number of comparisons, please ensure that the p value threshold for significance is adjusted for multiple comparisons.
4. Please address the potential relevance of these results to reports that Ras-pathway mutations confer resistance to venetoclax in AML patients.
Author Response
Dear Reviewer,
Thank you for your comments. We have added our responses under each comment, and highlighted red in the manuscript
- Reference 1 seems to be about AML it is associated in the introduction with a statement about ALL. Perhaps a more relevant reference can be cited.
New reference has been replaced which describes AML incidence in both ALL and AML.
Kuhlen M, Klusmann J-H, Hoell JI. Molecular Approaches to Treating Pediatric Leukemias. Front Pediatr [Internet]. 2019;7.
- For Figure 2A, please state if the values in the heatmap are normalized to a control value, and if so, what the control condition was.
This has been amended in the current draft- highlighted in red, please refer to line 245
(Values were normalized to DMSO controls).
- For the proteomics analysis, since there are a large number of comparisons, please ensure that the p value threshold for significance is adjusted for multiple comparisons.
This has been amended in the current draft- highlighted in red, please refer to line 365-366.
(p-values adjusted for multiple comparisons of < 0.05 was considered statistically significant).
- Please address the potential relevance of these results to reports that Ras-pathway mutations confer resistance to venetoclax in AML patients.
This has been discussed in the current draft- highlighted in red, please refer to line 484-487.
Studies have shown that AMLs harboring RAS-MAPK pathway-activating mutations (N/KRAS, FLT3, PTPN11) demonstrate resistance to Venetoclax-based therapies, and a recent study demonstrated the benefit of combinations therapies of Venetoclax with trametinib for RAS-mutated relapsed or refractory myeloid malignancies (35).
Reviewer 2 Report
Pediatric acute myeloid leukemia (AML) represents 15%–20% of all pediatric acute leukemias. Survival rates have increased over the past few decades to due to improved supportive care, optimized risk stratification and intensified chemotherapy. In most children, AML presents as a de novo entity. Diagnostic classification of pediatric AML includes a combination of morphology, cytochemistry, immunophenotyping and molecular genetics. Outcome is mainly dependent on the initial response to treatment and molecular and cytogenetic aberrations. Treatment consists of a combination of intensive anthracycline- and cytarabine-containing chemotherapy and stem cell transplantation in selected genetic high-risk cases or slow responders. In general, around 30% of all pediatric AML patients will suffer from relapse, whereas 5%–10% of the patients will die due to disease complications or the side-effects of the treatment. Targeted therapy may enhance anti-leukemic efficacy and minimize treatment-related morbidity and mortality, but requires detailed knowledge of the genetic abnormalities and aberrant pathways involved in leukemogenesis. Mylotarg is only approved targeted therapy for pediatric AML. Some of the Targeted therapies that are designed to induce apoptosis in AML. There are curently some of the most promising anti-leukaemia strategies, but using those therapies in combination will add a synergistic value by targeting multiple apoptotic targets. In this article, authors aim to investigate if targeting different apoptotic pathways through combinations may be more beneficial than single treatments. For that, they developed an algorithm with the capability of determining all possible pairwise combination for a given number or compounds. The article has minor points that authors need to reconsider:
1. The abstract is longer than Journal’s requirements of 200 words. Please simplify.
2. What are the keywords of the manuscript, please add 3-10 keywords following the abstract.
3. Lines 91-100 at page 2 and lines 101-103 at page 3 are describing the methodology of the study rather than introduction, please add these to the methods section.
4. Please add references to the studies mentioned in page 3 line 109-110 and page 3 lines 27-135.
5. Please add statistics to figure 3A.
6. In figure 3C, statistics show the comparison of which values? Please clarify.
7. Figure 7C is small to read, please make it bigger.
8. Please discuss the potential side effects of the triple combination
9. Did you work on real patient samples? Please add if any
Author Response
Dear Reviewer,
Thank you for your comments. We have added our responses under each comment, and highlighted red in the manuscript
- The abstract is longer than Journal’s requirements of 200 words. Please simplify.
This has been amended in the current draft to only 250 words.
- What are the keywords of the manuscript, please add 3-10 keywords following the abstract.
This has been amended in the current draft under the abstract
Keywords: AML, Apoptosis, Paediatric, Drug screening, Synergism, Double combination, Triple combination, Phosphoproteomics
- Lines 91-100 at page 2 and lines 101-103 at page 3 are describing the methodology of the study rather than introduction, please add these to the methods section.
This has been removed in the current draft. Please refer to the current manuscript.
- Please add references to the studies mentioned in page 3 line 109-110 and page 3 lines 27-135.
Those particular results were not published in our first paper by Lappin et al (2020), instead we decided to further investigate them through this study by Ali et al.
This has been clarified in the current draft to (data not published) and highlighted red in the manuscript. Please refer to line 136 and 156.
- Please add statistics to figure 3A.
This has been addressed in the current draft to include error bars.
- In figure 3C, statistics show the comparison of which values? Please clarify.
This has been amended in the current draft, please refer to line 327-328.
* = p < 0.05, ** = p < 0.005 values of significance of treatments compared to controls were calculated using paired t test.
- Figure 7C is small to read, please make it bigger.
This has been amended in the current draft to bigger size font.
- Please discuss the potential side effects of the triple combination
The study involved screening a library of apoptotic compounds that are still in the pre-clinical phase, therefore no data exists on possible side effects of triple combination. However, it is well known that the use of drugs in combination can produce a synergistic effect if each of the drugs impinges on a different target or signalling pathway that results in reduction of required drug concentrations for each individual drug, which will minimize any side effects of the monotherapy.
- Did you work on real patient samples? Please add if any
The present study is a proof of concept study to investigate synergism for combination therapies. Therefore we didn’t include any patient samples from our biobank, but we totally agree that a good validation for the current study would be validating the results in real patient samples, we appreciate this feedback and will consider it in our coming study.
Reviewer 3 Report
Ijms-2211266
Combination Therapies Targeting Apoptosis in Paediatric AML Understanding the molecular mechanisms of AML treatments using phosphoproteomics.
The article “Combination Therapies Targeting Apoptosis in Paediatric AML Understanding the molecular mechanisms of AML treatments using phosphoproteomics. (ijms-2211266)” by Ali A, et al. demonstrated that Bcl-2 inhibitor, CDK inhibitor and AKT inhibitor showed synergistic therapeutic effect for pediatric AML cell lines. This therapeutic effect was independent from caspase8 pathway, so called death ligand dependent apoptosis and dependent of Bcl-2 inhibition. In addition, therapeutic activity via Bcl-2 inhibition was regulated by not PP2A but ERK. This research article was very interesting. Therefore, I considered that this research article was suitable for acceptance of IJMS. However, I have several comments to improve this article.
1. This research article was described in detail, but the volume of main document was too heavy and the number of figures was too many for readers to understand well. Therefore, I considered that the authors could reduce the volume of manuscript.
2. The authors analyzed whether synergistic effect was shown or not using the combination index methods. Generally, the combination index was used for investigation in two therapeutics, but the authors investigated the activity of three drugs combination. Therefore, the authors explained how to analyze the combination index for three drugs in detail. Was that second step manner? For instance, first, BCL-2 inhibitor and AKT inhibitor worked synergistically, and second, CDK inhibitor and combination of BCL-2 inhibitor and AKT inhibitor worked synergistically.
3. In this study, Purvalanol-A and SU9516 were selected as CDK inhibitors. These two drugs were CDK 1, 2, and 4 inhibitors. Therefore, the readers could not understand what kinds of CDK inhibition was a key for therapeutic effect for pediatric AML. The author could add the experiment or discussion about that if possible.
Author Response
Dear Reviewer,
Thank you for your comments. We have added our responses under each comment, and highlighted red in the manuscript
- This research article was described in detail, but the volume of main document was too heavy and the number of figures was too many for readers to understand well. Therefore, I considered that the authors could reduce the volume of manuscript.
We do appreciate this feedback and are aware that the paper is described in detail, but we had to explain everything in detail especially we had conflicting data about the role of Bcl-2 phosphorylation in enhanced synergism of combination treatments. Therefore detailed description of mechanisms involved was necessary to understand this conjecture.
When possible we did cut parts of the manuscript including:
- Line 113-124 in the original manuscript.
- Line 154-157 in the original manuscript.
- Line 263-268 in the original manuscript.
- Line 276-277 in the original manuscript.
- Line 378-379 in the original manuscript.
- Line 421-425 in the original manuscript.
- The authors analyzed whether synergistic effect was shown or not using the combination index methods. Generally, the combination index was used for investigation in two therapeutics, but the authors investigated the activity of three drugs combination. Therefore, the authors explained how to analyze the combination index for three drugs in detail. Was that second step manner? For instance, first, BCL-2 inhibitor and AKT inhibitor worked synergistically, and second, CDK inhibitor and combination of BCL-2 inhibitor and AKT inhibitor worked synergistically.
In line 229-232 we explained that synergism was determined by using Compusyn software that was based on method used by Chou and Talalay. This allows for multiple comparison between drugs either double or triple. The Combination Index value – a dimensionless quantity for the determination and quantification of the drug interaction’s type – for the triple combinations was calculated automatically based on the general equation of the Combination Index at x% inhibition as follows:
Where, (Dx)1 is the dose of the drug D1 alone that inhibits the growth of cells by x%, (Dx)2 is the dose of the drug D2 alone that inhibits the growth of cells by x% and (Dx)3 is the dose of the drug D3 alone that inhibits the growth of cells by x%. (D)1, (D)2 and (D)3 are the doses of the drugs in combination that inhibit the growth of cells by x%. The (Dx)1, (Dx)2 and (Dx)3 values can be calculated from equation (4). If the CI value is equal to 1, additive effect is achieved. If the CI value is smaller than 1, synergistic interaction is achieved. If the CI value is greater than 1, the interaction type is antagonism.
- In this study, Purvalanol-A and SU9516 were selected as CDK inhibitors. These two drugs were CDK 1, 2, and 4 inhibitors. Therefore, the readers could not understand what kinds of CDK inhibition was a key for therapeutic effect for pediatric AML. The author could add the experiment or discussion about that if possible.
This has been clarified in the current draft, and highlighted red in the manuscript. Please refer to line 304-307.
Interestingly two apoptotic compounds in our combination therapies are CKD inhibitors -Purvalanol-A (CDK 2,4,9) and SU9516 (CDK 2).